# Freshness Analysis of Raw Laver (*Pyropia yenzoensis*) Conserved under Supercooling Conditions

**DOI:** 10.3390/foods12030510

**Published:** 2023-01-22

**Authors:** Hyeonbo Lee, Dong Hyeon Park, Eun Jeong Kim, Mi-Jung Choi

**Affiliations:** 1Department of Food Science and Biotechnology of Animal Resources, Konkuk University, Seoul 05029, Republic of Korea; 2Kimchi Industry Promotion Division, Practical Technology Research Group, World Institute of Kimchi, Gwangju 61755, Republic of Korea; 3Refrigerator Research of Engineering Division, Home Appliance and Air Solution Company, LG Electronics, Changwon 51533, Republic of Korea

**Keywords:** red seaweed, low-temperature preservation, algae, stepwise temperature algorithm, supercooling

## Abstract

Freezing raw laver is unsuitable for the laver industry due to process characteristics and economic problems. Therefore, this study attempted to investigate supercooled storage to extend the storage period without freezing, rather than refrigeration. To compare and analyze the storage ability of supercooling, the experiment was performed under refrigeration (5 °C), constant supercooling (CS, −2 °C), stepwise supercooling (SS, −2 °C), and freezing (−18 °C) conditions for 15 days, and the physicochemical changes according to the treatment and period were investigated. All SS samples, which were designed for stable supercooling, were kept in a supercooled state for 15 days. Two samples among the twelve total subjected to CS were frozen. At 9 days, the drip losses of the CS and SS samples were 6.32% and 6.48%, respectively, which was two times lower than that of refrigeration and three times lower than that of the frozen samples. The VBN of the refrigerated samples was 108.33 mg/100 g at 6 days, which exceeded the decomposition criterion. Simultaneously, the VBN of the other treatments was under the decomposition criterion of 30 mg/100 g. However, the VBN of both supercooling samples at 15 days increased to higher than the decomposition criterion. Regarding appearance, the refrigerated samples showed tissue destruction at 9 days, but tissue destruction of the CS and CC samples was observed at 15 days, and tissue destruction of the frozen samples was not observed until 15 days. Consequently, supercooling did not maintain quality for longer periods than freezing, but it did extend the shelf life more than refrigeration, and effectively preserved the quality for a short period.

## 1. Introduction

Raw laver (*Pyropia yenzoensis*) is the most widely farmed and consumed seafood in Northeast Asia [1]. It has been reported that this seaweed is commonly processed as dried sheets with various nicknames, such as ‘kim’ (Korea), ‘nori’ (Japan), and ‘zicai’ (China). Raw laver contains many functional ingredients, including polysaccharides, porphyrins, proteins, fatty acids, carotenoids, vitamins, and minerals, with particularly high protein (30–40%) and carbohydrate (40%) contents [2]. Although the demand for processed laver is continuously increasing, its harvest period is limited to approximately five months, from December to April. In addition, seaweed harvesting involves setting sail, and it is difficult to have a constant supply because of weather factors [3]. Furthermore, raw laver is prone to discoloration and decay after harvest; therefore, it is rapidly processed into dried laver on the day after reaping [4]. In conclusion, we rewrote this one because the intended meaning was not maintained., and this limited raw material distribution limits production planning and affects overall yields and sales [5]. Consequently, a short-term storage method is required to improve productivity. Raw laver is known to be resistant to freezing; as it is alive even in polar regions [6], it was thought that frozen storage methods would be advantageous, but freezing the bulk of raw laver is an inappropriate method for short-term storage followed by continuous processing. To overcome this issue, it is necessary to develop storage methods that can extend the shelf life while maintaining the quality of raw laver.

In food preservation, low-temperature storage by employing refrigeration and freezing is important for maintaining freshness [7]. However, refrigeration only extends the shelf life by a few days, and freezing causes quality degradation owing to the formation and growth of ice crystals [8,9]. Moreover, freezing is difficult to apply in the raw laver industry because of the large amount of energy required for freezing and thawing. To overcome this issue, it is necessary to develop novel preservation methods that extend the shelf life of raw lavers. In this regard, supercooling technology has emerged as an alternative preservation method, where the food product temperature is lowered below its freezing point without ice crystal formation [10]. Although supercooling prevents ice crystal formation, it is thermodynamically unstable, and ice nucleation can occur spontaneously [11]. Therefore, magnetic and electric fields, high pressure, and ultrasound waves have been applied with supercooling storage to improve the stability. Unfortunately, the effects of these techniques on supercooling are still dubious and limited because the underlying mechanism of preventing ice nucleation is not clear [7,12].

Several publications have reported the application of strict temperature control for the stable supercooling of various foods [13], and stepwise supercooling has significant advantages for preventing ice nucleation [7]. Fukuma et al. [14] used a stepwise cooling algorithm for the supercooling preservation of various fish samples. The samples were stored in incubators, with a decrease in temperature of 0.5 °C every 24 h, and were maintained under supercooled conditions until reaching −5 °C. Park et al. [9] successfully conducted a supercooling storage experiment using a stepwise cooling algorithm for mackerel. The temperature was set to reduce by 0.5 °C twice every 18 h, which was repeated every 72 h. However, experiments related to supercooling have been focused on maintaining the supercooling degree and for long-term supercooling storage [14,15]. In addition, most of the studies on raw laver are related to biomass raw material [16] or the freezing effects on photosynthesis [17], with few reports on food materials.

Therefore, this study was performed to analyze and compare the physicochemical properties of raw laver stored under refrigeration (5 °C), supercooling (−2 °C), and freezing (−18 °C) conditions. The supercooling preservation was divided into two treatments, with one constantly controlled at −2 °C and the other receiving an application of a stepwise cooling algorithm to maintain stable supercooling.

## 2. Materials and Methods

### 2.1. Sample Preparation

Fresh raw laver (*Pyropia yenzoensis*) was received as free material support from the Gyeonggi southern fisheries cooperatives (Hwaseong-si, Gyeonggi-do, Republic of Korea). The raw laver was packaged as 85 g samples in a polystyrene tray. A total of 51 packed samples were prepared and divided into 16 groups (4 storage periods and 4 types of storage in triplicate). The remaining samples were used as controls and not subjected to preservation treatment. Three packed samples were preserved in a refrigerator, and this process was performed using three devices.

### 2.2. Storage Methods

The packed samples were divided into four types of preservation treatments, and their physicochemical properties were analyzed after 3, 6, 9, and 15 days. The refrigeration treatment was performed in a refrigerator (R-F87HBSW, LG Electronics, Seoul, Korea) set to 5 °C. The freezing points of red algae such as *Pyropia yenzoensis* are −7/−8 °C for slow cooling, but the freezing point of the external seawater between the raw laver surface and layered blades is known to be about −2 °C [18]. Considering this point, the supercooling target temperature was set to −2 °C to avoid the release of supercooling during the storage period. The supercooling treatments were performed in refrigerators (K417SS13, LG Electronics) using two methods: maintaining at −2 °C and controlling by a 36 h cycle stepwise cooling algorithm. The initial temperature was set to −1 °C and was reduced twice by 0.5 °C every 12 h, which was repeated every 36 h. The freezing treatment was conducted in a freezer (R-F87HBSW, LG Electronics) set to −18 °C. To measure the temperature changes during storage, T-type thermocouples were attached to the packed sample surfaces and connected to a data logger (MX-100 Data Acquisition, Yokogawa, Tokyo, Japan). All frozen samples were thawed in a refrigerator (R-F87HBSW, LG Electronics) at 3 °C until the center temperature of the sample reached 0 °C 12 h before the experiment.

### 2.3. Drip Loss

Fresh raw lavers were weighed before packaging, and stored raw lavers were weighed after removing exudates from the dry tissues. The drip loss was calculated as the difference in the masses of the fresh and stored samples using the following formula:(1)$$\mathrm{Drip}\;\mathrm{loss}\;=\left\{\left({\;W}_{1}-W_{2}\right)/W_{1}\right\}\times 100$$
where W_1_ is the initial weight of the sample (g) and W_2_ is the weight of the treated sample (g).

### 2.4. Moisture Content

The moisture content of the samples was determined using an oven-drying method according to AOAC 930.04 [19]. The samples were dried at 105 °C and weighed again until a constant weight was achieved. The moisture content was calculated using the following formula:(2)$$\mathrm{Moisture}\;\mathrm{content}(\%)=\{(W_{1}-W_{2})/W_{1}\}\times 100$$
where W_1_ is the sample weight before drying (g) and W_2_ is the sample weight after drying (g).

### 2.5. pH

The pH of the raw laver was measured using a modified method of Lee et al. [11]. The sample pH was measured in triplicate using a pH meter (S-220, Mettler Toledo Co., Zurich, Switzerland). Approximately 25 g of the treated sample was homogenized with 250 mL of distilled water for 60 s using a hand blender (4199, De’Longhi Romania SRL, Juc-Herghelie Romania).

### 2.6. Thiobarbituric Acid Reactive Substance (TBARS)

The TBARS assay was performed in accordance with the method described by Park et al. [9]. Approximately 0.2 g of the sample was homogenized in 5 mL of 0.1% trichloroacetic acid (TCA), and the homogenate was subsequently centrifuged at 10,000 rpm for 5 min at 4 °C. For every l mL of supernatant, 4 mL of 20% TCA containing 0.5% thiobarbituric acid (TBA) was added. After heating at 95 °C for 30 min, the mixture was cooled in an ice bath and centrifuged at 10,000 rpm for 15 min at 4 °C. The absorbance of the supernatant was measured using a spectrophotometer (MultiskanTM GO UV/VIS, Thermo Fisher, Massachusetts, USA) at 535 nm and corrected for non-specific turbidity by subtracting the absorbance at 600 nm. The TBARS contents of the samples are expressed as mg of malondialdehyde/kg (mg MDA/kg).

### 2.7. Volatile Basic Nitrogen (VBN)

The VBN content of the raw laver samples was determined according to Conway’s microdiffusion method [9]. First, 5 g of the sample was homogenized in 45 mL of distilled water using a slap-type homogenizer (WS-400, Shanghai Zhisun Equipment, Shanghai, China) for 180 s. The homogenate was then filtered using Whatman No. 1 filter paper (GE Healthcare Life Sciences, Sheffield, UK). After filtering, 1 mL of the filtrate was placed in the outer section of a Conway dish with 1 mL of 0.01 N H_3_BO_3_, and 100 μL of Conway solution (0.066% methyl red and 0.066% bromocresol green in ethanol) was dropped into the inner section of the dish. Next, 1 mL of 50% K_2_CO_3_ was added to the outer section of the dish, which was then closed and incubated at 37 °C for 2 h. Afterward, the mixture was titrated using 0.02 N H_2_SO_4_ until the Conway reagent changed to a red color. The VBN content was determined following the addition of 0.02 N H_2_SO_4_ to the inner section of the Conway dish, and the VBN values were calculated using the following formula:(3)$$\mathrm{VBN}\;\left(\mathrm{mg}\;/\;100\;g\right)=\frac{14.007\times \left(a-b\right)\times \;f\;\times 100\times \;c}{S}$$
where a is the titration volume of the sample (mL); b is the titration volume of the blank (mL); f is the factor for H_2_SO_4_; S is the sample weight (g); and c is the dilution amount.

### 2.8. Total Aerobic Count (TAC)

First, 20 mL of sterilized plate count agar (PCA; Plate Count Agar, Difco, NJ Franklin Lakes, USA) medium was added into a petri dish and left to harden. Thereafter, 1 mL of the sample and 1 mL of 1/10 dilution aliquots (1 mL) were plated in duplicate on PCA medium and incubated for 48 h at 37 °C, after which the media with 30–300 colony-forming units (CFUs) were counted. Duplicate values for each of the three treatment replicates were averaged, and the data are reported as log CFU/mL. The triplicate values were averaged to obtain the data reported herein.

### 2.9. Color

The sample color was measured after removing moisture from the surface. For each sample, the lightness (Commission Internationale de l’Eclairag (CIE) L*), redness (CIE a*), and yellowness (CIE b*) were measured using a chromameter (CR-400, Minolta Co., Ltd., Osaka, Japan). The instruments were calibrated using standard white tiles prior to the analysis, and the measurements were repeated 5 times. The ΔE value, representing the color difference, was calculated using the following equation:(4)$$\Delta E=\sqrt{{\left({\Delta \mathrm{CIE}\;L}^{*}\right)}^{2}+{\left({\Delta \mathrm{CIE}\;a}^{*}\right)}^{2}+{\left({\Delta \mathrm{CIE}\;b}^{*}\right)}^{2}}$$

The visible appearance was recorded using a light box (Photo Studio Light Box Portable, PULUZ, Shenzhen, China) and a digital camera (SONY A7II, SONY, Tokyo, Japan).

### 2.10. Pigment Contents

The photosynthetic pigment (chlorophyll a) was extracted from milled freeze-dried laver samples (0.1 g) in 5 mL of dimethyl sulfoxide (DMSO) for 12 h in the dark. The extracts were centrifuged at 4500× *g* at 30 °C for 15 min. The supernatant absorbance was measured at 664 and 648 nm, and the concentration of the photosynthetic pigment was calculated according to the following equation [20]:(5)$$\mathrm{Chlorophyll}\;a\;\left(\mu g/\mathrm{mL}\right)=12.25\times A_{664\mathrm{nm}}-2.79\times A_{648\mathrm{nm}}$$

Phycoerythrin (PE) was extracted from milled, freeze-dried laver samples. The ground samples (0.1 g) were diluted in 5 mL of 0.1 mol/L buffer phosphate solution (pH 6.8) and extracted for 12 h in the dark. The extracts were centrifuged at 4500 rpm at 4 °C for 15 min. The supernatant absorbance was measured at 455, 564, and 592 nm, and the concentration of the photosynthetic pigment was calculated according to the following equation [21]:(6)$$\mathrm{PE}\;\left(\mathrm{mg}/g\right)=\left[\left(A_{564\mathrm{nm}}-A_{592\mathrm{nm}}\right)-\left(A_{455\mathrm{nm}}-A_{592\mathrm{nm}}\right)\times 0.20\right]\times 0.12$$

### 2.11. Sodium Dodecyl Sulfate-Polyacrylamide Gel Electrophoresis (SDS-PAGE)

The SDS-PAGE of raw laver was conducted using a modified method of Shin et al. [22]. The soluble protein for SDS-PAGE was extracted from the milled freeze-dried laver samples (0.25 g) suspended in deionized water (10 mL). The suspension was stirred at 150 rpm overnight at 4 °C. After incubation, the suspension was centrifuged at 10,000 rpm for 1 h. The supernatant was filtered and prepared for SDS-PAGE via lysing, denaturing, and reducing using Laemmli sample buffer (Bio-Rad, Hercules, CA, USA) and β—mercaptoethanol (Bio-Rad, Hercules, CA, USA) at 90 °C for 5 min. Subsequently, 20 μL of the sample was loaded into a 12% acrylamide precast tris-glycine gel (Bio-Rad, Hercules, CA, USA) for SDS-PAGE separation using a Mini-Protean Tetra Cell unit (Bio-Rad, Hercules, CA, USA). The gels were run at a stacking voltage of 100 V until the samples ran out of the wells, followed by a run at a constant voltage of 200 V. The loaded gel was stained with 0.02% (*w*/*v*) Coomassie brilliant blue R250 (B7920; Sigma, USA) in 50% (*v*/*v*) methanol and 7.5% (*v*/*v*) acetic acid. The gel was destained with destaining solution 1, containing 40% methanol and 7% acetic acid, for 30 min and destained overnight in destaining solution 2, containing 15% methanol and 7% acetic acid. The separated protein bands were compared with standard protein markers (Precision Plus Protein Standards, Bio-Rad, Hercules, CA, USA).

### 2.12. Free Amino Acid Content

Free amino acids were extracted from the freeze—dried laver samples (0.2 g) in 10 mL of a 3% TCA solution and extracted at room temperature (25 °C) for 1 h. The extracts were then centrifuged at 15,000 rpm at room temperature for 15 min. The supernatants were filtered through a 0.45 µm membrane filter (Millipore Corp., Bedford, MA, USA) and analyzed using an amino acid analyzer (S433, Sykam GmbH, Eresing, Germany). The instrumental parameters were as follows: column temperature −60 °C; injection volume—10 μL; wavelength detection—440 and 570 nm; and flow rate—0.4 mL/min.

### 2.13. Statistical Analysis

All experiments were performed in triplicate, and the average values are reported. The statistical analysis was performed using the SPSS Statistics version 24.0 software for Windows (Statistical Package for the Social Science, Ver. 24.0 IBM., Chicago, IL, USA). To confirm significant differences among treatments, a one-way analysis of variance (ANOVA) and Duncan’s multiple range test were performed at the 95% confidence level (*p* < 0.05).

## 3. Results and Discussion

### 3.1. Time–Temperature Curves

The time–temperature curves of the raw laver samples over 15 days of storage are shown in Figure 1. Raw laver stored under refrigerated conditions reached 5 °C after 1.6 h of storage. The supercooling treatments were divided into two cooling methods. The constant supercooling treatment involved placing the samples in a refrigerator maintained at −2 °C, and two samples among twelve total were frozen for 15 days. A stepwise cooling algorithm was adopted for stable supercooling preservation. This algorithm was designed with the target temperature of −2.0 °C, and the primary temperature was set at −1.0 °C and declined by 0.5 °C every 12 h. The temperature was maintained after attaining −2.0 °C for 12 h. After that, the temperature was increased to −1.0 °C to prevent ice nucleation and then decreased by 0.5 °C every 12 h. This program was cycled every 36 h to prevent ice nucleation. During the stepwise cooling algorithm, the sample temperatures followed the target temperature within ± 0.2 °C. The sample temperatures were stably sustained in the range of −2.2 to −1.8 °C without a phase transition, and all samples were unfrozen during the storage periods. The mean target temperatures of the supercooling treatments were −2 and −1.5 °C, which are close to the freezing point of seafood [13]. Therefore, the use of a stepwise cooling algorithm for supercooling preservation was successful. The freezing treatment was performed in a refrigerator set to −18 °C, and it required approximately 7 h for the samples to equilibrate to −18 °C. The frozen samples were stored at −16 to −20 °C for 15 days.

### 3.2. Drip Loss, Moisture Content, and pH

#### 3.2.1. Drip Loss

The drip loss of raw laver stored under various treatment conditions and at different periods is presented in Figure 2a. Drip loss is an important factor for food quality and it affects microbial growth [11]. After 9 days of storage, the drip loss of the refrigerated samples was not significantly different (*p* < 0.05). Afterward, the refrigerated sample drip loss abruptly increased and was the highest among those of all treatments at 15 days (*p* < 0.05), which is consistent with the notable increase in the drip loss from day 8 when *Palmaria palmata* and *Gracilaria tikvahiae*, other red algae, were preserved at 2 °C and 7 °C [23]. At 15 days, the drip losses of the supercooled samples increased abruptly, showing no significant differences compared to those of the freezing treatment (*p* > 0.05). Figure 3a shows the decomposed blade tissue of the refrigeration samples and both supercooled sample groups (SC and CC) after 9 and 15 days, respectively. Based on these results, it is suspected that the loss of cellular liquid and internal materials was due to the destruction of laver blade tissue. The frozen samples showed the highest drip loss after 6 days of storage (*p* < 0.05), but it slowly increased compared to that of the refrigerated samples. It is likely that water in the cells formed ice crystals, which damaged the permeation barrier of the cells, allowing internal materials to escape after freezing and thawing [8,9]. The drip loss of the CS and SS samples at 9 days was 6.32% and 6.48%, respectively, which was two times lower than the refrigeration samples and three times lower than the frozen samples (*p* < 0.05). This result, in which the drip loss of the supercooled samples was significantly lower than that of the refrigeration and freezing samples, was the same for pork belly, chicken breast, and mackerel fillets [7,9]. Supercooled preservation does not produce ice crystals as with freezing, and the tissue destruction is slower than with refrigeration [8]. Therefore, supercooling preservation is advantageous for reducing the drip loss during storage.

#### 3.2.2. Moisture Content

The moisture content values of the raw laver samples subjected to various storage treatments and periods are shown in Figure 2b. The moisture content of the fresh raw laver was 87.05%, which decreased with different storage treatments and periods. Generally, the respiration and transpiration of algae proceeded after harvesting, reducing the water content [24]. The moisture content of the frozen samples decreased more rapidly than that of the samples subjected to other treatments, showing the lowest moisture value after 15 days of storage (*p* < 0.05). This water loss occurred because the cells were damaged by ice crystals generated during freezing, allowing intracellular water to escape [25]. The moisture content of the CS samples showed significantly lower values than that of the control group from day 3 of storage (*p* < 0.05) and demonstrated a lower moisture content than that of the refrigeration samples during the storage period. In contrast, the moisture content of the refrigerated samples at 6 days and the SS samples at 9 days showed no significant difference when compared with that of fresh samples, being maintained at 86% or higher (*p* > 0.05). According to Jo and Surh [26], when food is stored at a low temperature, the difference in the water vapor pressure between the food and the storage environment increases, resulting in increased moisture loss. Therefore, the storage temperature of the CS samples, which was lower than the average preservation temperature of the refrigeration and SS samples, could have caused more moisture loss. On the other hand, the moisture content of low-temperature-preserved sweet cherries showed different results than raw laver, where the 0 °C sample had a higher moisture content than the 4 °C sample for 20 days. This result was due to respiration and transpiration, which occurred slower under the lower temperature, affecting the moisture loss of sweet cherries [27]. As a result, it could be considered that the effect of metabolic inhibition and the difference in water vapor pressure on the moisture content of raw seaweed is different depending on the storage temperature.

#### 3.2.3. pH

The raw laver pH was measured after preservation using various storage methods and periods, and the results are shown in Figure 2c. The pH of fresh raw laver was 6.53, which is very close to that of other marine species such as fish, mollusks, and shellfish, and even other edible algae species [28]. The refrigerated samples showed rapid increases in pH after 6 days of storage. The increased pH during storage resulted from the formation of basic compounds generated from protein degradation by microorganisms and enzymes [29]. This result was attributed to the protein that is abundant in *Pyropia yenzoensis* (25–50%) and decomposition, which increased the basic compound content in the matrix [11,30]. The supercooled sample pH showed different trends compared to those of the refrigerated samples. Overall, the pH decreased to < 6.25, which was significantly lower than that of the control (*p* < 0.05). In particular, the pH of the CC samples was the lowest among those of all treated samples (*p* < 0.05), likely due to organic acid formation owing to the metabolic activity of the stored samples and microorganisms [11]. The frozen sample pH showed a pattern similar to that of the supercooled samples. Therefore, supercooling is more beneficial for maintaining the pH of fresh raw laver compared to refrigeration storage.

### 3.3. Color and Pigment Contents

#### 3.3.1. Appearance and Color

The appearance and color of raw laver stored under different conditions and periods are presented in Figure 3a–e. Generally, consumers consider color an important factor in choosing fresh and dried laver [31]. High-quality laver has a black color, whereas discolored red laver has no commercial value [32]. The CIE L* value of the fresh samples was 21.83, which was affected by the storage temperature. The CIE L* of the refrigerated samples increased more than that of the other samples during the storage period. However, the CIE L* values of the supercooled samples were significantly lower than those of the refrigerated samples (*p* < 0.05). The CIE a* values of the treated samples generally increased, and the refrigeration treatment induced a significant increase compared with the other treatment conditions. In particular, the color change of the refrigerated samples was more noticeable than that of the other treated samples (Figure 3a), indicating a reduced commercial value [32]. Furthermore, the refrigerated samples showed more tissue destruction than the samples subjected to other treatments after 15 days of storage. The CIE b* value of the refrigerated samples was 3.10 at 15 days of storage, which is the highest among those of all treatment groups (*p* < 0.05). The total color difference (ΔE) of the refrigerated raw laver increased rapidly after 9 days of storage, reaching its highest value after 15 days of storage (*p* < 0.05). Meanwhile, the ΔE change during the storage of *Palmaria palmata* and *Gracilaria tikvahiae*, which are red algae species along with raw laver (*Pyropia yenzoensis*), showed different aspects. The ΔE of *Palmaria palmata* increased significantly from the 7th day at 7 °C, and no significant difference was observed until 10 days at 2 °C. In *Gracilaria tikvahiae*, the ΔE showed no significant change at both 2 °C and 7 °C for 12 days [23]. According to Takahashi et al. [33], laver discoloration is caused by pigment synthesis inhibition owing to low temperatures, which is inconsistent with the observations of this study. This was expected due to the different environments between cultivation and post-harvest preservation. In a storage environment where nitrogen nutrient sources are limited, it is expected that a low temperature would suppress metabolism and delay the exhaustion of nutrient sources to suppress discoloration [23]. In addition, it is thought to be the result of various variables other than gene expression acting at low temperatures. Considering these results, supercooling preservation is more beneficial for maintaining the fresh raw laver color compared to refrigeration.

#### 3.3.2. Pigment Content

The pigment contents of raw laver stored under various conditions and periods are shown in Figure 3f and g. The chlorophyll-a (Chl-a) and PE of fresh raw laver were 1.28 µg/g and 1.52 mg/g, respectively, with the two values showing opposite trends. Regarding the chl-a content (Figure 3f), the chl-a concentration of the frozen samples was not significantly different during storage (*p* > 0.05). Likewise, the chl-a concentrations of both the SC and CC samples were not significantly different for 9 days (*p* > 0.05). However, that of the refrigerated samples significantly decreased on day 9th (*p* < 0.05). Some authors have reported that chl-a in various plants is more stably preserved at lower temperatures (up to −20 °C), even through freezing [34,35]. The degradation of pigments in the chloroplast is caused by various causes, such as reduced oxidation, enzymatic reactions, temperature, or by some other factors or combinations thereof [33]. The lower the temperature, the more restrained the enzymatic reaction that degrades thylakoid membrane-binding proteins and pigments [35]. At 15 days, both supercooling samples showed the lowest chl-a content (*p* < 0.05). The pH could have caused these results. The pH decrease could have initiated proteolytic enzyme activity, which degrades the thylakoid membrane-binding proteins for chlorophyll [36]. PE is a more stable protein than chl-a [37]. In addition, even if PE is decomposed, color develops if the chromophore is not decomposed [38]. These characteristics caused PE to show higher values than fresh samples at 15 days, unlike chl-a. Frozen samples showed the fewest changes in PE, and there was no significant difference when compared with fresh ones at 15 days (*p* > 0.05). However, the PE of the refrigeration samples significantly increased from the 3rd day (*p* < 0.05). The PE of both supercooled samples was significantly lower than that of the refrigeration samples during storage (*p* < 0.05). The SC samples showed a higher PE than the CC samples (*p* < 0.05). This was judged to be the effect of the storage temperature and the storage algorithm. These results were expected due to the low-temperature stress response of *Pyropia yenzoensis*. It is known to repress the synthesis of PE with gene expression at low temperatures [39]. Consequently, supercooling is more advantageous than refrigeration for preventing pigment degradation.

### 3.4. TBARS, VBN, and TAC

#### 3.4.1. TBARS

The TBARS values of the raw laver samples subjected to various storage treatments and periods are shown in Figure 4a. The flavor is an important quality factor when customers buy dried laver, which is primarily processed raw laver [40]. Lipid oxidation, related to a rancid odor and discoloration, is caused by active oxygen and hydroxyl radicals from respiration and photosynthesis [41]. In addition, lipid oxidation is temperature-dependent and effectively inhibited by low-temperature storage [9,11]. The TBARS value of fresh raw laver was 0.21 mM MDA/g and was influenced by the storage temperature. The TBARS values of the refrigerated samples were significantly higher than those of the other samples after 6 days (*p* < 0.05). Afterward, the TBARS content of the refrigerated samples rapidly increased to 0.99 mM MDA/g at 15 days, reaching the highest value among those of all treated samples (*p* < 0.05). On the 9th day, the frozen samples had a value of 0.23 mM MDA/g, and the SS and CS samples had a value of 0.22 mM MDA/g, showing no significant difference from the fresh samples (*p* > 0.05). On the 15th day, the SC, CC, and frozen samples showed significantly higher values than the fresh samples (*p* < 0.05). However, the SS samples had a value of 0.32 mM MDA/g, the CS samples had a value of 0.28 mM MDA/g, and the frozen samples had a value of 0.27 mM MDA/g, which was more than three times lower than that of the refrigerated samples on the same day. Therefore, supercooling and freezing were more effective in preventing lipid oxidation than refrigeration.

#### 3.4.2. VBN

VBN is a critical parameter for protein-containing foods, as it can be used to evaluate volatile compound contents [42]. The VBN contents of raw laver subjected to various storage treatments and periods are shown in Figure 4b. The VBN value of fresh raw laver was 9.33 mg/100 g, which was differently affected by the storage methods and periods used herein. The VBN value of the refrigerated samples rapidly increased during the storage period and was the highest among those of all treated samples (*p* < 0.05). This showed a similar aspect to the pH trend of refrigerated samples (Figure 2c). After 6 days of storage, the VBN value of the refrigerated samples exceeded 30 mg/100 g, which satisfied the criteria for decomposition [42]. However, the VBN values in the other treatments were < 30 mg/100 g, and those of the supercooled samples slowly increased over 9 days of storage without significant differences between the supercooling treatment types. Subsequently, the VBN value of the supercooled samples abruptly increased to > 280 mg/100 g, but that of the SS samples was lower than that of the SC samples (*p* < 0.05). This was a different trend from the pH of both supercooling samples, where the lowest value was observed at 15 days. This result demonstrates that supercooling better suppresses the basic compound content in the matrix than refrigeration [9]. In contrast, the VBN values of the frozen samples were significantly unchanged during the storage period (*p* < 0.05). These results showed the same trend as the results of preserved mackerel fillets for 12 days under refrigeration, supercooling (−2 °C), and freezing, but the VBN of raw laver demonstrated a higher value than that of the mackerel fillets. The mackerel fillets had a value of less than 16 mg/100 g for 12 days, even for the refrigerated samples, and supercooling and freezing exhibited highly fresh conditions (7.88–8.68 mg/100 g) [9]. According to Park et al. [7], proteins are degraded by ammonia and amines via the proteolytic activities of enzymes and bacteria. Therefore, low-temperature storage is necessary to inhibit protein degradation, and supercooling was determined to be superior to refrigeration for delaying raw laver degradation.

#### 3.4.3. TAC

The TAC of raw laver samples subjected to various treatments and storage periods is shown in Figure 3c. According to Lee et al. [11], microbial activity affects the shelf life of food and is temperature-dependent. The TAC of the fresh raw laver samples was 3.28 log CFU/g, which is similar to the value obtained by Jeong et al. [43]. After 6 days of storage, the TAC of the refrigerated samples was 4.25 log CFU/g, which is significantly higher than those of the samples stored under the other conditions (*p* < 0.05). However, the frozen samples showed a TAC value of 3.05 log CFU/g on day 15, which was significantly lower than those of the refrigerated and supercooled samples (*p* < 0.05). The TAC values of the supercooled samples were lower than those of the refrigerated samples after 6 days of storage, but were higher than those of the frozen samples (*p* < 0.05). Ye et al. [44] reported that low-temperature storage is essential for reducing microbial growth, and the results presented herein indicate that supercooling is more effective for limiting microorganisms in raw laver than refrigeration.

### 3.5. SDS-PAGE

To measure changes in the protein content, the SDS-PAGE results of raw laver stored under different conditions and periods are presented in Figure 5. Raw laver contains phycobiliprotein combined with pigment protein, accounting for 20% of its weight [45]. Overall, the protein degradation trend was similar to that of the VBN results. According to the authors of [46], phycobiliprotein is composed of an αβ heterodimer and γ with a molecular weight of 13–33 kDa. During storage, the 50 kDa band of the refrigerated samples disappeared faster than that of the other treated samples. No bands were detected at < 15 kDa, and the supercooled samples showed similar proteolytic patterns after 6 days of storage. However, during the SS treatment, which had a relatively higher storage temperature than the CS treatment, the protein decomposed into small molecules after 9 days of storage. However, the band of the frozen samples remained constant and was similar to that of the fresh samples. Based on these results, supercooling preservation is more effective for delaying protein degradation in raw laver compared to refrigeration.

### 3.6. Free Amino Acid Concentration

The free amino acid (FFA) contents of the raw laver stored under various conditions and periods are listed in Table 1 The composition and content of FAAs are important indicators of flavor [47]. According to Kawashima et al. [48], the FAAs that impart sweet and umami flavors include taurine, alanine, glutamic acid, aspartic acid, glycine, proline, serine, and threonine. Glutamic acid is the most important amino acid for the production of umami [49]. The most abundant amino acid in fresh laver was taurine (12.37 mg/g), followed by alanine (6.86 mg/g), glutamic acid (3.73 mg/g), and aspartic acid (1.07 mg/g) in descending order. These FAAs accounted for approximately 92% of the total content, and after 9 days of storage, the taurine content of the treated samples increased compared to that of fresh samples, but then decreased to < 11 mg/g, except for the samples subjected to the freezing treatment. The frozen sample alanine content was also higher than that of the fresh raw laver. The aspartic acid concentration in all treated samples decreased compared to that in the fresh samples after 3 days of storage. The aspartic acid content of the refrigerated samples decreased more rapidly than that of the other samples. The glutamic acid content showed similar trends to that of aspartic acid. However, the glutamic acid content of the supercooled samples was higher than that of the refrigerated samples. Glutamate ammonia ligase catalyzes glutamine synthesis from ammonia and glutamic acid [49]. This likely contributed to the rapidly decreasing glutamic acid content of the refrigerated samples compared to those of the other treated samples. In contrast, a greater amount of ammonia was generated in the refrigerated samples than in the other samples, as the protein was microbially decomposed in a process significantly affected by the storage temperature [11], resulting in a high VBN value. After 15 days of storage, α−AAA was detected in the refrigerated and SC samples, which was likely produced via protein oxidation due to the relatively high storage temperatures [49]. Based on these results, supercooling preservation is more beneficial than refrigeration for suppressing amino acid degradation, as lower average storage temperatures resulted in improved food quality parameters.

## 4. Conclusions

This study attempted to investigate supercooling raw laver (*Pyropia yenzoensis*) to extend the preservation period without freezing. The stepwise cooling algorithm was applied to prevent ice nucleation. The SS treatment was able to preserve a more stable supercooling than the CS treatment by the algorithm. The SS and CS samples showed less drip loss than the frozen samples until 9 days. The frozen samples showed the closest values to fresh samples among all conditions for the TBARS, VBN, TAC, total color difference, and pigment content. In both supercooling conditions, the VBN value satisfied the criteria for decomposition until day 6, and the TBARS did not show a significant difference from freezing for 15 days (*p* > 0.05). Furthermore, the SS and CS samples exhibited superior TBARS, VBN, and TAC values compared to the refrigerated samples. The color change and pigment content of the refrigerated samples were significant compared to those of the other treatments. Protein degradation in the raw laver was slower under supercooled conditions compared to that of refrigeration. The SS and CS samples showed no significant differences in their drip loss, pH, TBARS, TAC, and chl-a concentrations during the storage period (*p* > 0.05). However, the SC samples, which were exposed to a higher average temperature than the CS samples, showed a significantly higher total color difference than the CC samples on the 6th day, and the PE concentration also showed a significantly higher value than the CS samples on the 3rd day (*p* < 0.05). Consequently, the results presented herein suggest that supercooling can extend the shelf life of raw laver as a novel low-temperature storage method. In addition, further research is needed to increase the supercooling stability of raw laver at a lower temperature than this study.

## Figures and Tables

**Figure 1 foods-12-00510-f001:**
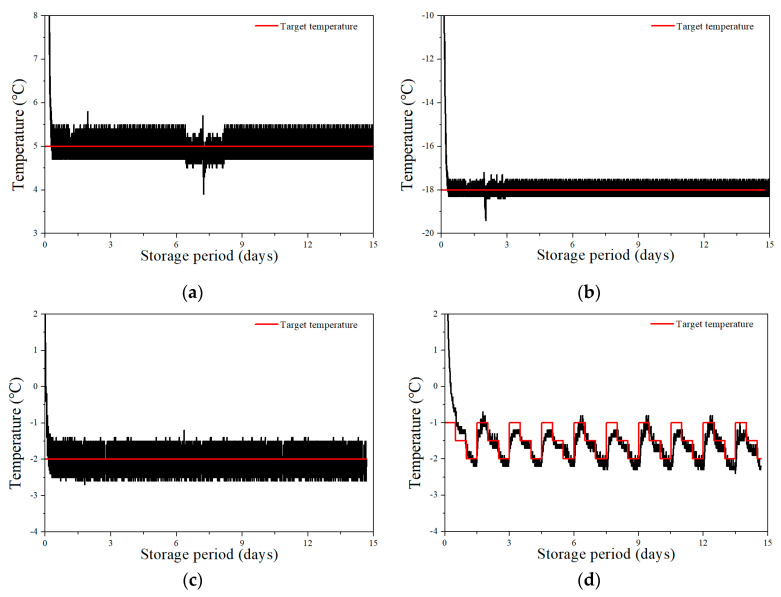
Time-temperature profiles of the target temperature and raw laver samples stored under the conditions of (**a**) refrigeration, (**b**) freezing, (**c**) constant-temperature supercooling, and (**d**) supercooling using a stepwise cooling algorithm.

**Figure 2 foods-12-00510-f002:**
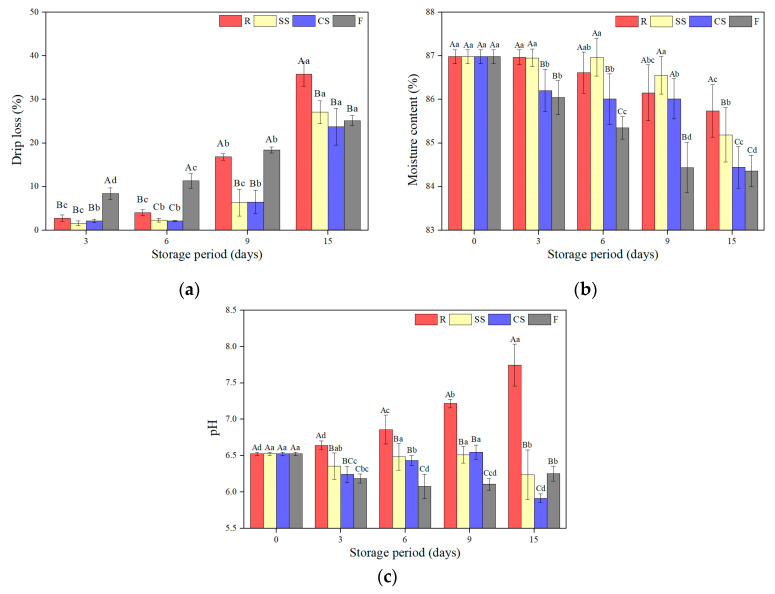
Changes in (**a**) drip loss, (**b**) moisture content, and (**c**) pH of raw laver (*Pyropia yenzoensis*) depending on various storage methods and periods. R, refrigeration; SS, supercooling by stepwise cooling algorithm; CS, constant temperature supercooling; F, freezing. ^A–C^ Means with different letters within the same storage period are significantly different (*p* < 0.05). ^a–d^ Means with different letters within the same storage temperature are significantly different (*p* < 0.05).

**Figure 3 foods-12-00510-f003:**
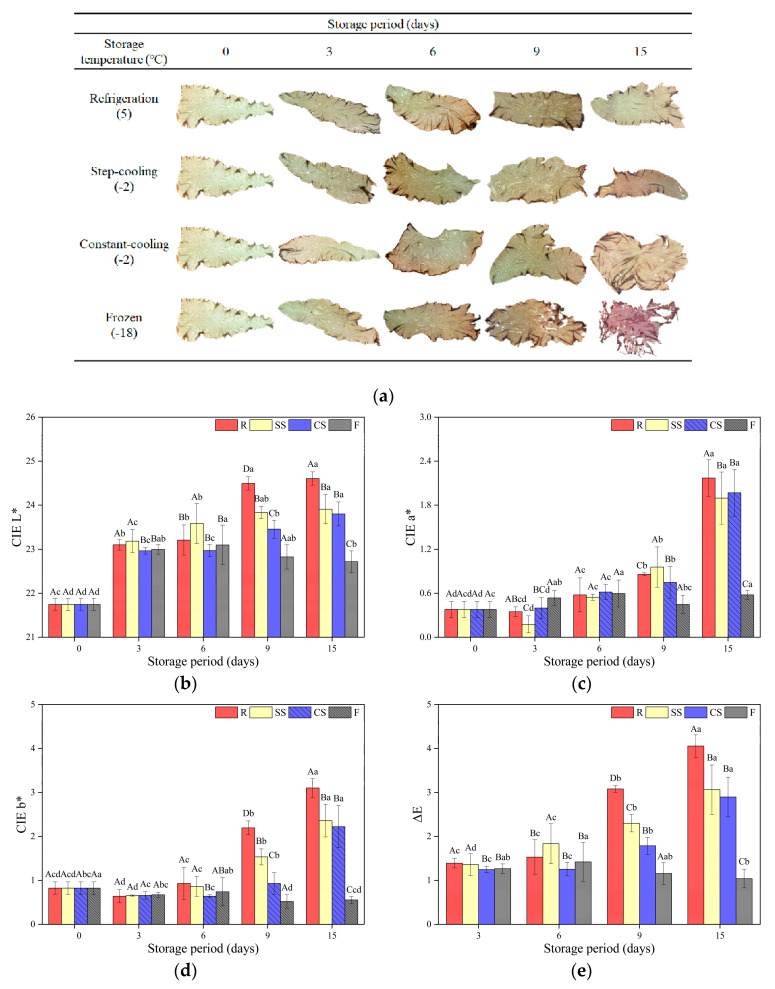
Changes in (**a**) the appearance of laver, (**b**) CIE L*, (**c**) CIE a*, (**d**) CIE b*, (**e**) total color difference, (**f**) chlorophyll-a (**Chl-a**) and (**g**) phycoerythrin (**PE**) of raw laver (*Pyropia yenzoensis*) depending on various storage methods and periods. R, refrigeration; SS, supercooling by stepwise cooling algorithm; CS, constant temperature supercooling; F, freezing. ^A–D^ Means with different letters within the same storage period are significantly different (*p* < 0.05). ^a–d^ Means with different letters within the same storage temperature are significantly different (*p* < 0.05).

**Figure 4 foods-12-00510-f004:**
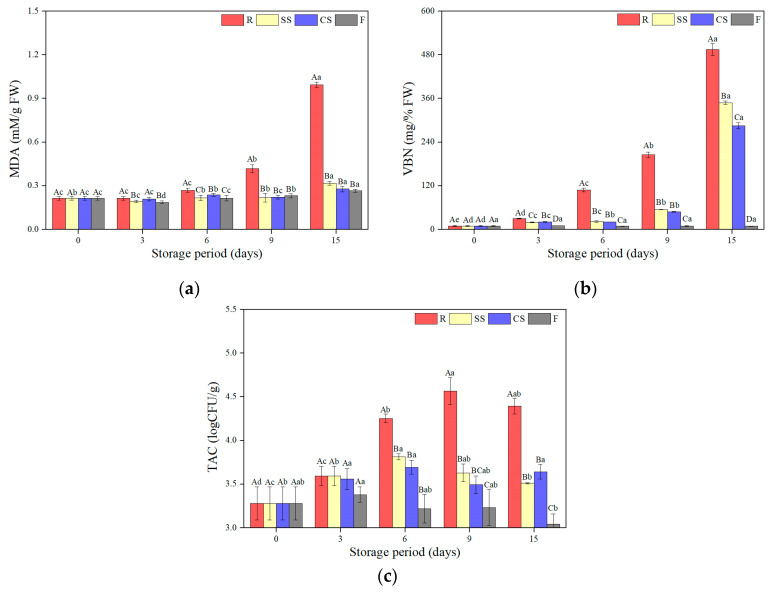
Changes in (**a**) thiobarbituric acid reactive substances (TBARS), (**b**) volatile basic nitrogen (VBN), and (**c**) total aerobic bacteria count (TAC) of raw laver (*Pyropia yenzoensis*) depending on various storage methods and periods. R, refrigeration; SS, supercooling by stepwise cooling algorithm; **CS**, constant temperature supercooling; F, freezing. ^A–D^ Means with different letters within the same storage period are significantly different (*p* < 0.05). ^a–d^ Means with different letters within the same storage temperature are significantly different (*p* < 0.05).

**Figure 5 foods-12-00510-f005:**
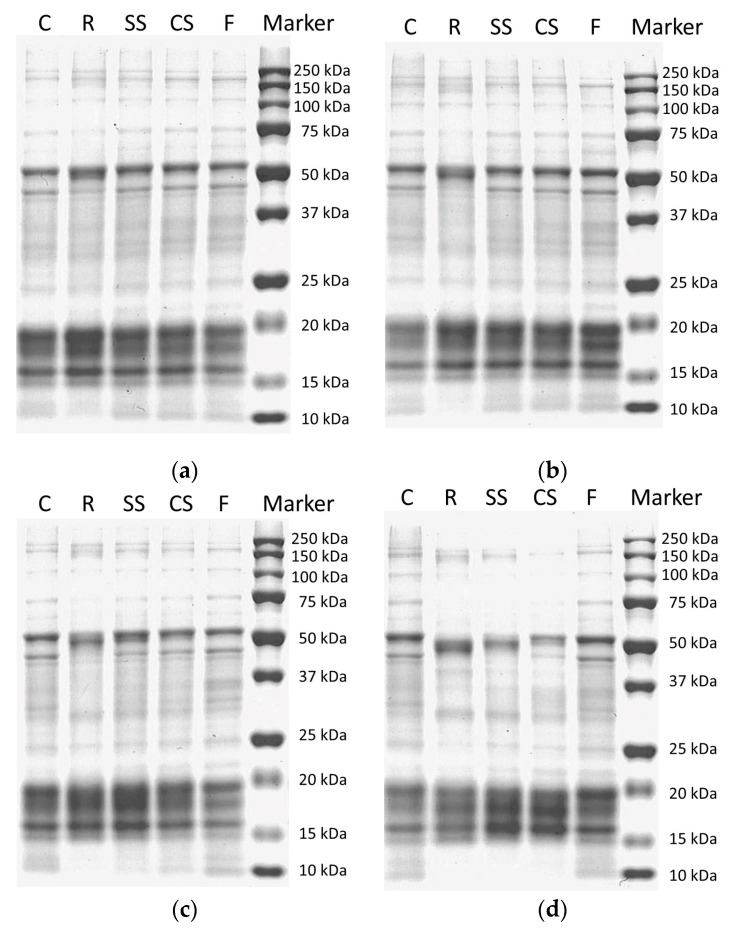
Sodium dodecyl sulfate-polyacrylamide gel electrophoresis (SDS-PAGE) of raw laver (*Pyropia yenzoensis*) depending on various storage methods and periods: (**a**) 3 days, (**b**) 6 days, (**c**) 9 days, and (**d**) 15 days. C, fresh sample; R, refrigeration; SS, supercooled by stepwise cooling algorithm; CS, constant temperature supercooling; F, freezing.

**Table 1 foods-12-00510-t001:** Free amino acid content of raw laver (*Pyropia yenzoensis*) depending on various storage methods (refrigeration, supercooling with stepwise cooling algorithm, constant temperature supercooling, and freezing) for 15 days.

Free Amino Acid Content (mg/g)						
Free Amino Acid	Storage Method	Storage Period (Day)				
		0	3	6	9	15
Tau	R	12.37 ± 0.01	12.92 ± 0.24	12.57 ± 0.10	12.41 ± 0.23	10.24 ± 0.09
	SS		12.96 ± 0.11	12.25 ± 0.10	12.54 ± 0.10	10.01 ± 0.09
	CS		12.18 ± 0.20	12.23 ± 0.23	12.23 ± 0.21	10.90 ± 0.13
	F		13.73 ± 0.20	13.31 ± 0.12	12.94 ± 0.20	13.06 ± 0.18
Ala	R	6.86 ± 0.01	8.44 ± 0.06	8.79 ± 0.02	8.59 ± 0.03	5.65 ± 0.02
	SS		8.35 ± 0.05	8.51 ± 0.05	8.71 ± 0.05	4.72 ± 0.03
	CS		7.40 ± 0.05	8.99 ± 0.08	9.01 ± 0.04	7.22 ± 0.06
	F		8.60 ± 0.03	8.69 ± 0.06	8.45 ± 0.05	9.18 ± 0.02
Glu	R	3.73 ± 0.01	2.92 ± 0.01	2.74 ± 0.01	1.93 ± 0.01	0.34 ± 0.01
	SS		3.31 ± 0.02	2.99 ± 0.02	1.74 ± 0.01	0.46 ± 0.01
	CS		2.99 ± 0.02	3.49 ± 0.02	2.25 ± 0.02	1.02 ± 0.01
	F		3.97 ± 0.01	3.17 ± 0.03	4.17 ± 0.02	3.23 ± 0.03
Asp	R	1.07 ± 0.01	0.90 ± 0.01	0.70 ± 0.01	0.49 ± 0.02	0.05 ± 0.01
	SS		1.01 ± 0.01	0.78 ± 0.03	0.53 ± 0.01	0.12 ± 0.03
	CS		0.84 ± 0.01	0.83 ± 0.04	0.62 ± 0.01	0.17 ± 0.03
	F		0.77 ± 0.01	0.69 ± 0.01	0.67 ± 0.01	1.34 ± 0.07
Cit	R	0.54	0.81 ± 0.08	0.67 ± 0.04	0.36 ± 0.04	0.18 ± 0.02
	SS		0.72 ± 0.08	0.73 ± 0.02	0.60 ± 0.04	0.05
	CS		0.72 ± 0.04	0.71 ± 0.07	0.62 ± 0.03	0.27 ± 0.02
	F		0.76 ± 0.06	0.57 ± 0.07	0.62 ± 0.03	0.81 ± 0.05
Thr	R	0.25 ± 0.01	0.48 ± 0.03	0.47 ± 0.03	0.36 ± 0.01	0.02 ± 0.01
	SS		0.36 ± 0.03	0.34 ± 0.03	0.30 ± 0.02	0.12 ± 0.04
	CS		0.30 ± 0.02	0.34 ± 0.03	0.28 ± 0.06	0.14 ± 0.02
	F		0.32 ± 0.02	0.27 ± 0.04	0.29 ± 0.02	0.38 ± 0.06
g-ABA	R	0.25 ± 0.01	0.35 ± 0.04	0.41 ± 0.02	0.39 ± 0.03	0.10 ± 0.02
	SS		0.40 ± 0.03	0.41 ± 0.05	0.40 ± 0.03	0.13 ± 0.06
	CS		0.38 ± 0.03	0.37 ± 0.03	0.47 ± 0.02	0.29 ± 0.04
	F		0.35 ± 0.03	0.29 ± 0.05	0.32 ± 0.02	0.39 ± 0.04
Ser	R	0.24 ± 0.01	0.18 ± 0.03	0.19 ± 0.03	0.09 ± 0.01	0.01
	SS		0.16 ± 0.02	0.17 ± 0.02	0.11 ± 0.03	0.07
	CS		0.14 ± 0.02	0.16 ± 0.03	0.11 ± 0.03	0.06
	F		0.14 ± 0.02	0.11 ± 0.01	0.13 ± 0.02	0.15 ± 0.01
Val	R	0.22 ± 0.01	0.47 ± 0.05	0.47 ± 0.02	0.50 ± 0.04	1.15 ± 0.09
	SS		0.39 ± 0.06	0.44 ± 0.04	0.39 ± 0.06	0.40 ± 0.02
	CS		0.35 ± 0.08	0.46 ± 0.03	0.41 ± 0.02	0.34 ± 0.02
	F		0.42 ± 0.02	0.33 ± 0.04	0.32 ± 0.07	0.50 ± 0.02
Leu	R	0.17 ± 0.01	0.50 ± 0.02	0.50 ± 0.01	0.48 ± 0.03	0.88 ± 0.01
	SS		0.35 ± 0.01	0.41 ± 0.02	0.37 ± 0.01	0.32 ± 0.02
	CS		0.31 ± 0.01	0.42 ± 0.01	0.34 ± 0.01	0.26 ± 0.02
	F		0.35 ± 0.01	0.25 ± 0.02	0.26 ± 0.03	0.43 ± 0.01
P—Ser	R	0.17 ± 0.01	0.27 ± 0.02	0.32 ± 0.03	0.33 ± 0.02	0.88 ± 0.04
	SS		0.20 ± 0.01	0.19 ± 0.01	0.23 ± 0.01	0.28 ± 0.01
	CS		0.16 ± 0.02	0.20 ± 0.01	0.24 ± 0.01	0.31 ± 0.01
	F		0.18 ± 0.01	0.20 ± 0.03	0.20 ± 0.01	0.21 ± 0.01
b—Ala	R	0.13 ± 0.01	0.14 ± 0.01	0.15 ± 0.01	0.15 ± 0.02	0.12 ± 0.01
	SS		0.14 ± 0.01	0.14 ± 0.01	0.15 ± 0.01	0.10 ± 0.01
	CS		0.14 ± 0.01	0.13 ± 0.01	0.14 ± 0.01	0.10 ± 0.01
	F		0.13 ± 0.01	0.12 ± 0.01	0.13 ± 0.01	0.14 ± 0.01
Ile	R	0.11 ± 0.01	0.31 ± 0.03	0.30 ± 0.01	0.27 ± 0.01	0.47 ± 0.01
	SS		0.22 ± 0.01	0.24 ± 0.02	0.21 ± 0.01	0.15 ± 0.01
	CS		0.18 ± 0.01	0.25 ± 0.01	0.21 ± 0.01	0.16 ± 0.01
	F		0.21 ± 0.02	0.15 ± 0.01	0.16 ± 0.01	0.25 ± 0.01
Lys	R	0.11 ± 0.01	0.30 ± 0.02	0.29 ± 0.01	0.29 ± 0.04	0.01 ± 0.01
	SS		0.23 ± 0.01	0.27 ± 0.01	0.23 ± 0.01	0.29 ± 0.01
	CS		0.20 ± 0.01	0.30 ± 0.02	0.24 ± 0.02	0.19 ± 0.02
	F		0.24 ± 0.01	0.17 ± 0.01	0.17 ± 0.01	0.30 ± 0.03
Tyr	R	0.09 ± 0.01	0.18 ± 0.03	0.19 ± 0.02	0.19 ± 0.03	0.57 ± 0.04
	SS		0.15 ± 0.01	0.18 ± 0.02	0.14 ± 0.01	0.12 ± 0.01
	CS		0.13 ± 0.01	0.18 ± 0.02	0.14 ± 0.01	0.10 ± 0.01
	F		0.17 ± 0.01	0.13 ± 0.01	0.13 ± 0.01	0.21 ± 0.02
Gly	R	0.09 ± 0.01	0.12 ± 0.01	0.19 ± 0.01	0.54 ± 0.05	1.15 ± 0.05
	SS		0.14 ± 0.01	0.18 ± 0.01	0.32 ± 0.01	0.92 ± 0.03
	CS		0.11 ± 0.01	0.19 ± 0.01	0.25 ± 0.01	0.44 ± 0.02
	F		0.17 ± 0.01	0.13 ± 0.01	0.14 ± 0.01	0.16 ± 0.02
Pro	R	0.06 ± 0.01	0.21 ± 0.02	0.18 ± 0.03	0.14 ± 0.01	ND
	SS		0.12 ± 0.01	0.17 ± 0.03	0.15 ± 0.02	ND
	CS		0.10 ± 0.01	0.23 ± 0.04	0.12 ± 0.01	ND
	F		0.19 ± 0.02	0.14 ± 0.01	0.11 ± 0.01	0.17 ± 0.04
Phe	R	0.06 ± 0.01	0.17 ± 0.01	0.15 ± 0.01	0.16 ± 0.01	0.53 ± 0.01
	SS		0.13 ± 0.01	0.16 ± 0.01	0.13 ± 0.01	0.13 ± 0.01
	CS		0.12 ± 0.01	0.17 ± 0.01	0.13 ± 0.01	0.14 ± 0.01
	F		0.16 ± 0.01	0.12 ± 0.01	0.12 ± 0.01	0.19 ± 0.01
Arg	R	0.04 ± 0.01	0.04 ± 0.01	0.03 ± 0.01	0.03 ± 0.01	0.01 ± 0.01
	SS		0.06 ± 0.01	0.05 ± 0.01	0.04 ± 0.01	0.03 ± 0.01
	CS		0.07 ± 0.01	0.09 ± 0.01	0.06 ± 0.01	0.15 ± 0.01
	F		0.09 ± 0.01	0.10 ± 0.01	0.10 ± 0.02	0.09 ± 0.01
NH3	R	0.03 ± 0.01	0.12 ± 0.01	0.70 ± 0.05	1.28 ± 0.05	2.80 ± 0.10
	SS		0.04 ± 0.01	0.06 ± 0.01	0.15 ± 0.01	1.10 ± 0.03
	CS		0.03 ± 0.01	0.04 ± 0.01	0.07 ± 0.01	0.56 ± 0.01
	F		0.04 ± 0.01	0.03 ± 0.01	0.03 ± 0.01	0.05 ± 0.01
His	R	0.03 ± 0.01	0.06 ± 0.02	0.06 ± 0.01	0.05 ± 0.01	0.02 ± 0.01
	SS		0.05 ± 0.01	0.06 ± 0.01	0.04 ± 0.01	0.04 ± 0.01
	CS		0.04 ± 0.01	0.06 ± 0.01	0.05 ± 0.01	0.03 ± 0.01
	F		0.05 ± 0.01	0.04 ± 0.01	0.04 ± 0.01	0.07 ± 0.01
a—ABA	R	0.02 ± 0.01	0.07 ± 0.01	0.08 ± 0.02	0.10 ± 0.02	0.59 ± 0.01
	SS		0.05 ± 0.01	0.04 ± 0.01	0.04 ± 0.01	ND
	CS		0.04 ± 0.01	0.05 ± 0.01	0.04 ± 0.01	ND
	F		0.04 ± 0.01	0.00 ± 0.01	0.04 ± 0.01	0.05 ± 0.01
Orn	R	0.01	0.07 ± 0.02	0.19 ± 0.01	0.27 ± 0.03	0.45 ± 0.04
	SS		0.03 ± 0.01	0.03 ± 0.01	0.03 ± 0.01	0.27 ± 0.01
	CS		0.03 ± 0.01	0.04 ± 0.01	0.03 ± 0.01	0.15 ± 0.01
	F		0.04 ± 0.01	0.03 ± 0.01	0.03 ± 0.01	0.03 ± 0.01
Met	R	0.01 ± 0.01	0.01	0.01	0.01	0.02
	SS		0.01	0.01	0.01	0.01
	CS		0.01	0.02	0.01	0.04
	F		0.02	0.02	0.02	0.01
a—AAA	R	ND	ND	ND	ND	0.08 ± 0.01
	SS		ND	ND	ND	0.04 ± 0.01
	CS		ND	ND	ND	ND
	F		ND	ND	ND	ND
Total content	R	26.67 ± 0.26	30.05 ± 0.77	30.34 ± 1.12	29.40 ± 0.78	26.31 ± 0.55
	SS		29.60 ± 1.23	28.81 ± 0.64	27.56 ± 0.66	19.87 ± 0.23
	CS		26.98 ± 0.54	29.93 ± 0.40	28.07 ± 0.32	23.03 ± 0.43
	F		31.14 ± 0.35	29.06 ± 0.22	29.57 ± 1.14	31.37 ± 1.01

R, refrigeration; SS, supercooled by stepwise cooling algorithm; CS, constant temperature supercooling; F, freezing. Standard deviation values are not displayed because they are less than *p* < 0.01. ND: not detected.

## Data Availability

The data are included in the article.

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
