# Peer review of "Freshness Analysis of Raw Laver (*Pyropia yenzoensis*) Conserved under Supercooling Conditions"

_foods, 2023, doi:10.3390/foods12030510_

Round 1
Reviewer 1 Report
The manuscript presents high quality data and results presented in the research.
The text is well written, with high technical quality of relevant essays and analyzes to elucidate how best to store and conserve the raw laver (Pyropia yenzoensis).
I found Figure 4(a) very relevant, which shows the images of the leaves throughout the treatment. This adds a lot to the reader in viewing the results in practice, in addition to the quantitative data from the analyses.
I have only one doubt regarding the need for TBARs analysis, since vegetables and leafy greens in general are low in fat. Moreover, the results showed low values of TBARs in all treatments, being only relevant in the 15-day refrigerated treatment, which was expected.
Author Response
- I understand your points, completely. As you have mentioned TBARS which represents lipid oxidation could not critical due to the low fat content in raw laver. However, this study is the first research to analyze raw laver (Pyropia yenzoensis) by low temperature preservation from the point of food, and the flavor is traditionally an important factor in the quality of laver. Therefore, it was intended to provide information on lipid rancidity, which is one of the factors that cause rancid odor. We appreciated your points. Considering your point, we modified the TBARS section and added other cite to explain its validity.
Page 9. Line 343-348: The TBARS values of the raw laver samples subjected to various storage treatments and periods are shown in Figure 3a. The flavor is an important quality factor when customers buy dried laver which is primarily processed raw laver [31]. Lipid oxidation related to rancid odor and discoloration is caused by active oxygen and hydroxyl radicals from respiration and photosynthesis [32]. In addition, Lipid oxidation is temperature dependent and effectively inhibited by low temperature storage [9, 11].

Reviewer 2 Report
Article describes quality analyzes of an interesting food, with effective methodological basis. However some changes are needed in the context for future publication. Consideration should be given to some improvements in writing and formatting of the text in addition to questionable points, as described in sequence.
Article needs a detailed review of English for adequate language adjustment.
Summary: The information presented is not clear. Objective has not been defined. Furthermore, no conclusions were presented. I suggest redoing it, presenting the main objective, which analyzes were performed (in addition to the treatments that the samples were submitted), the main findings and the conclusion (which is the best evaluated treatment).
I also suggest that adjustments be made to the Keywords. should look for descriptor terms different from those presented in the title.
Line 61. I could cite the main limitations, to better support the applied method.
Furthermore, make a selection of references, opting for the most current ones (last 5 years).
Line 84. When referring to "4 packages", it is understood that there were different types of packaging, but only the Styrofoam packaging is described. Rethink!
Lines 101-102. Times have already been described in that same section.
Line 111. "AOAC (2012)". Update and add the analysis number/code.
Session 2.5. Add method reference.
Line 227. Better detail supercooling time/temperature progression using a cooling algorithm.
Line 236. "Consistent with the findings of [22]". What is the tested product? Must detail.
Line 247-248. "Therefore, supercooled preservation..." "Therefore, supercooling preservation is..." Similarity that justifies the need for a detailed review of English.
Lines 305-306. Expression not consistent with the described findings (no significant differences among the storage treatments were observed p>0.05). Rewrite.
Throughout the discussion, there is a lack of comparisons between the values obtained with other studies carried out, in addition to the evaluated treatments.
Table 2. Title needs to present more details. In addition, all acronyms must be highlighted in the legend.
I also suggest that a similarity analysis be performed to verify that all citations were properly performed in the manuscript.
Author Response
Article describes quality analyzes of an interesting food, with effective methodological basis. However some changes are needed in the context for future publication. Consideration should be given to some improvements in writing and formatting of the text in addition to questionable points, as described in sequence.
Article needs a detailed review of English for adequate language adjustment.
- We appreciated your points. The language organization were revised by a professional and English-native experts before submission.
Summary: The information presented is not clear. Objective has not been defined. Furthermore, no conclusions were presented. I suggest redoing it, presenting the main objective, which analyzes were performed (in addition to the treatments that the samples were submitted), the main findings and the conclusion (which is the best evaluated treatment).
- Thank you for your critical points. As suggested, abstract section was rewritten about this point.
Page 1. Line 13-28: Freezing raw laver is unsuitable for the laver industry as process characteristics and economic problems. Therefore, this study attempted supercooling preservation to extend the storage period without freezing rather than refrigeration. To compare and analyze the storage ability of supercooled, the experiment was performed under refrigeration (5 °C), constant supercooling (CS, -2 °C), stepwise supercooling (SS, -2 °C), and freezing (-18 °C) conditions for 15 days, and physicochemical changes according to treatment and period were investigated. All SS samples which were designed for stable supercooling kept supercooled state for 15 days. Whereas two samples among twelve total subjected to CS were frozen. At 9 day, drip loss of CS and SS was 6.32% and 6.48%, respectively which is double lower than refrigeration and triple lower than frozen. VBN of refrigerated sample was 108.33 mg/100 g, which exceeded decomposition criteria at 6 days. However, VBN of both supercooling samples at 15 days increased over decomposition criterion. In appearance, the refrigeration showed tissue destruction at 9 days, but tissue destruction of CS and CC was observed at 15 days, and frozen samples were not observed until 15 days. Consequently, supercooling could not maintain quality for long periods than freezing, but it could extend shelf life than refrigeration, and effectively preserve quality for a short period.
I also suggest that adjustments be made to the Keywords. should look for descriptor terms different from those presented in the title.
- As suggested, we have amended the keywords to terms that differ from those presented in the title.
Page 1, line 43-45: Red seaweed, low-temperature preservation, algae, stepwise temperature algorithm, supercooling
Q1. Line 61. I could cite the main limitations, to better support the applied method.
Furthermore, make a selection of references, opting for the most current ones (last 5 years).
A1. As suggested, we added detailed limitations and the cited literature was altered within last 5 years.
Page 2, line 77-80: Unfortunately, the effects of these techniques on supercooling are still dubious and limited because the underlying mechanism of preventing ice nucleation is not made clear. [7, 12].
Page 18, line 595-596: Kang, T., You, Y., Jun, S. Supercooling preservation technology in food and biological samples: a review focused on electric and magnetic field applications. Food Sci. Biotechnol. 2020, 29, 303–321.
Q2. Line 84. When referring to "4 packages", it is understood that there were different types of packaging, but only the Styrofoam packaging is described. Rethink!
A2. We apologize for your confusion. we have been modified the word.
Page 3, line 102: 4 storage periods
Q3. Lines 101-102. Times have already been described in that same section.
A3. As suggested, we have removed redundant sentences and added necessary information.
Page 3, line 122: 12 hours before the experiment.
Q4. Line 111. "AOAC (2012)". Update and add the analysis number/code.
A4. As suggested, we updated and added the analysis number/code and modified a typo in that cite.
Page 3, line 133: 930.04,
Page 18, line 609: 19th ed.
Q5. Session 2.5. Add method reference.
A5. As suggested, We added method reference.
Page 3, line 139: The pH of raw laver was conducted using a modified method of Lee et al. [11].
Q6. Line 227. Better detail supercooling time/temperature progression using a cooling algorithm.
A6. As suggested, a detailed description of the stepwise cooling algorithm has been added to the text.
Page 6, line 237-242: This algorithm was designed with the target temperature of −2.0 °C, the initial temperature was set at −1.0 °C and declined by 0.5 °C every 12 h. The temperature was maintained after attaining −2.0 °C for 12 h. After that, the temperature was increased to −1.0°C to inhibit ice nucleation and then decreased by 0.5 °C every 12 h. This program was cycled every 36 h to prevent ice nucleation.
Q7. Line 236. "Consistent with the findings of [22]". What is the tested product? Must detail.
As suggested, the cited literature was altered to allow for more appropriate comparisons and we wrote the tested product.
Page 9. Line 355-364: Afterward, the refrigerated sample drip loss abruptly increased and was the highest among those for all treatments at 15 days (p<0.05), which is consistent with the increase notably in drip loss from day 8 when Palmaria palmata and Gracilaria tikvahiae, other red algae, were preserved at 2 °C and 7 °C. [23].
Page 18. Line 617-618: Nayyar, D.; Skonberg, D.I. Contrasting effects of two storage temperatures on the microbial, physicochemical, and sensory properties of two fresh red seaweeds, Palmaria palmata and Gracilaria tikvahiae. J. Appl. Phycol. 2019, 31, 731-739.
Q8. Line 247-248. "Therefore, supercooled preservation..." "Therefore, supercooling preservation is..." Similarity that justifies the need for a detailed review of English.
A8. Thank you for your critical points. As suggested, we removed unnecessary words.
Page 6, line 277-280: Supercooled preservation does not produce ice crystals like freezing, and tissue destruction is slower than refrigeration.
Q9. Lines 305-306. Expression not consistent with the described findings (no significant differences among the storage treatments were observed p>0.05). Rewrite.
A9. We appreciated the reviewer’s points. As suggested, we have rewritten the result description.
Page 9. Line 342-347: Afterward, the TBARS content of the refrigerated sample rapidly increased to 0.99 mM MDA/g at 15 days, reaching the highest value among those of all treated samples (p<0.05). On the 9th day, the frozen sample was 0.23 mM MDA/g, and the SS and CS were 0.22 mM MDA/g, showing no significant difference from the fresh sample (p>0.05). On the 15th day, SC, CC, and frozen samples showed significantly higher values than the fresh samples (p<0.05). However, SS was 0.32 mM MDA/g, CS mM MDA/g was 0.28 mM MDA/g, and the frozen sample was 0.27 mM MDA/g, which was more than triple times lower than the refrigerated sample on the same day. Therefore, supercooling and freezing were more effective in preventing lipid oxidation than refrigeration.
Throughout the discussion, there is a lack of comparisons between the values obtained with other studies carried out, in addition to the evaluated treatments.
- Thank you for your critical points. As suggested, we complemented the discussions in all manuscript.
Table 2. Title needs to present more details. In addition, all acronyms must be highlighted in the legend.
- As suggested, We have highlighted all acronyms in the legend in all manuscript. Title 2 was renamed Title 1 during the revision and corrected by adding more information to the title.
Page 15, line 531-533: Free amino acid content of raw laver (Pyropia yenzoensis) depending on various storage methods (refrigeration, supercooling with stepwise cooling algorithm, constant temperature supercooling, and freezing) for 15 days
I also suggest that a similarity analysis be performed to verify that all citations were properly performed in the manuscript.
As suggested, we executed similarity analysis of our manuscript with plagiarism checker (copykiller.com), and the similarity was 10%. In addition, we have attached a full text plagiarism analysis report.

Reviewer 3 Report
In the manuscript ‘Freshness analysis of raw laver (Pyropia yenzoensis) conserved under supercooling condition’, the authors attempted to use the supercooled storage method to extend the storage period. They investigated the quality of fresh laver under refrigeration (5 °C), constant supercooling (CS, -2 °C), stepwise supercooling (SS, -2 °C), and freezing (-18 °C) conditions afetr 15 days of storage. The quality change was analyzed by measuring several factors such as pH, TBARS, VBN, TAC etc. Generally, the experiment was straightforwardly designed and the manuscript was easy to understand. However, the scienfic value of the manuscript should be enhanced and major revisions be performed.
Specific comments:
1. Did the authors measure the freezing point of raw laver (Pyropia yenzoensis)? why did they set the supercooling temperature as -2 °C?
2. How to measure the freshness of raw laver scientifically? What are the major factors responsible for freshness? For quality assessment, I think sensory evaluation is quite important.
3. It is inappropriate to label the results in SDS-PAGE as Table 1. Why did the authors not identify the ezymes responsible for protein degradation?
Author Response
In the manuscript ‘Freshness analysis of raw laver (Pyropia yenzoensis) conserved under supercooling condition’, the authors attempted to use the supercooled storage method to extend the storage period. They investigated the quality of fresh laver under refrigeration (5 °C), constant supercooling (CS, -2 °C), stepwise supercooling (SS, -2 °C), and freezing (-18 °C) conditions afetr 15 days of storage. The quality change was analyzed by measuring several factors such as pH, TBARS, VBN, TAC etc. Generally, the experiment was straightforwardly designed and the manuscript was easy to understand. However, the scienfic value of the manuscript should be enhanced and major revisions be performed.
Specific comments:
Q1. Did the authors measure the freezing point of raw laver (Pyropia yenzoensis)? why did they set the supercooling temperature as -2 °C?
We did not measure the freezing point of raw laver. The freezing points of red intertidal seaweeds such as Pyropia yenzoensis. were reported −7/−8°C for slow cooling and −3.4/−4.5°C for rapid cooling (Pearson and Davison 1993). Meanwhile, the freezing point of seawater is approximate −2°C. It suggests that no internal ice nucleation occurs in raw laver at −2°C although external water is frozen. In this study, supercooling target temperature was set to -2 because it was necessary to track the freshness of the raw laver when supercooling proceeded stably.
In that section, the reasons for setting the supercooling temperature at -2 and related references have been added.
Page 3. Line 110-114: The freezing points of red seaweeds such as Pyropia yenzoensis were −7/−8°C for slow cooling, but the freezing point of external water which between the raw laver surface and layered blades, seawater, is known to be about -2 °C[18]. Considering this point, the supercooling target temperature was set to -2 °C to avoid the release of supercooling during the storage period.
Q2. How to measure the freshness of raw laver scientifically? What are the major factors responsible for freshness? For quality assessment, I think sensory evaluation is quite important.
A2. Thank you for your comments. Raw laver (Pyropia yenzoensis) is generally primarily processed immediately after harvest, it is difficult to find references measuring the freshness of raw laver. According to some references, the criterion for judging the freshness of a product customarily is the color change. Nutritionally, since the protein content of raw laver is very high (30-50%), protein deterioration could be cited as the main factor in quality change.
It is known that raw laver deteriorates quickly after harvest. Because of this characteristic, raw laver undergoes primary processing through heat drying immediately after harvest. Therefore, sensory evaluation was not conducted due to uncertainty about the stability of refrigerated raw laver.
Q3. It is inappropriate to label the results in SDS-PAGE as Table 1. Why did the authors not identify the ezymes responsible for protein degradation?
A3. We appreciated the reviewer’s points. Table 1 was modified to Figure 5 by referring to the reference cited in the experimental method of SDS-PAGE.
In this experiment, we tried to confirm the protein degradation degree of raw laver according to storage method and period. Therefore, we didn’t care about specific metabolic mechanisms such as identifying the enzymes responsible for protein degradation.

Round 2
Reviewer 2 Report
I congratulate the authors for the textual changes made in response to the requests sent in the first review. Such adjustments made the text more consistent for publication.
I suggest only brief correction for the final publication of the article: Abstract. description of the acronym VBN (volatile basic nitrogen).
Line 531: Delete (Pyropia yenzoensis).
Reviewer 3 Report
The authors have revised the manuscript according to the reviews' comments and it should be accepted for publication as it is.